# Combining Discrete and Continuous Representation: Scale-Arbitrary Super-Resolution for Satellite Images

**Tai An** [1,2] ⓘ**, Chunlei Huo** [1,2,3,]*****, Shiming Xiang** [1,2] ⓘ **and Chunhong Pan** [1]

[1] National Laboratory of Pattern Recognition, Institute of Automation, Chinese Academy of Sciences, Beijing 100190, China; antai2018@ia.ac.cn (T.A.)
[2] School of Artificial Intelligence, University of Chinese Academy of Sciences, Beijing 101408, China
[3] School of Automation and Electrical Engineering, University of Science and Technology, Beijing 100083, China
***** Correspondence: clhuo@nlpr.ia.ac.cn

**Abstract:** The advancements in image super-resolution technology have led to its widespread use in remote sensing applications. However, there is currently a lack of a general solution for the reconstruction of satellite images at arbitrary resolutions. The existing scale-arbitrary super-resolution methods are primarily predicated on learning either a discrete representation (DR) or a continuous representation (CR) of the image, with DR retaining the sensitivity to resolution and CR guaranteeing the generalization of the model. In this paper, we propose a novel image representation that combines the discrete and continuous representation, known as CDCR, which enables the extension of images to any desired resolution in a plug-and-play manner. CDCR consists of two components: a CR-based dense prediction that gathers more available information and a DR-based resolution-specific refinement that adjusts the predicted values of local pixels. Furthermore, we introduce a scale cumulative ascent (SCA) method, which enhances the performance of the dense prediction and improves the accuracy of the generated images at ultra-high magnifications. The efficacy and dependability of CDCR are substantiated by extensive experiments conducted on multiple remote sensing datasets, providing strong support for scenarios that require accurate images.

**Keywords:** scale-arbitrary super-resolution; image representation; satellite imagery

## 1. Introduction

Constrained by transmission bandwidth and hardware equipment, the spatial resolution of received remote sensing images may be inadequate, resulting in insufficient details and failing to meet the requirements of certain practical applications. Moreover, the variety of resolutions available at ground terminals makes it imperative to reconstruct satellite images at arbitrary scales. In real-world remote sensing applications, the ability to represent images at arbitrary resolutions is also crucial for object detection, semantic segmentation, mapping, and human–computer interaction.

Digital images are typically composed of discrete pixels, each of which represents different levels of detail at different scales. Single-image super-resolution (SISR) is a widely used computer vision technique that aims to reconstruct images at various scales. Thanks to the progress in deep learning, SISR models that operate on fixed integer scale factors (e.g., $\times 2/\times 3/\times 4$) have made significant advancements. However, most existing SISR models are limited to generating images with fixed integer scale factors, reducing their efficacy in remote sensing applications. Given the impracticality of training numerous models for multiple scale factors, developing a SISR method that can accommodate arbitrary (including non-integer) scale factors remains an open challenge.

In existing natural image-oriented, scale-arbitrary super-resolution techniques, two representative methods are Meta-SR [1] and LIIF [2]. Both methods make assumptions that each pixel value is composed of RGB channels. They predict the specific RGB values of each pixel in the high-resolution (HR) space based on the feature vector, also known

as the latent code, in the low-resolution (LR) space. However, their specific designs are different. On the one hand, the meta upscale module in Meta-SR generates convolution kernels with specific numbers and weights according to the scale factor. These kernels are then convolved with the latent code to predict the RGB value of a specific pixel. This approach of mapping the latent code to RGB values is referred to as discrete representation (DR). On the other hand, the local implicit image function (LIIF) directly predicts the RGB value of a pixel based on both the coordinates and the latent code. In contrast to the discrete point-to-point feature mapping in DR, LIIF creates the continuous representation (CR) of an image through continuous coordinates.

In comparison to discrete digital images, human perception of real-world scenes is continuous, thus both the discrete representation (DR), as represented by Meta-SR [1], and the continuous representation (CR), as represented by LIIF [2], can be utilized. CR employs a neural network-parameterized implicit function for continuous, global, and robust learning, while DR utilizes a multilayer perceptron (MLP) for discrete, local, and sensitive learning. In brief, CR enables reconstruction at ultra-high magnifications, while DR produces a more accurate image with sharper edges by adapting to specific resolutions. In this paper, we propose a novel method called combined discrete and continuous representation (CDCR) that incorporates the strengths of both CR and DR.

As illustrated in Figure 1, CDCR starts by producing a dense prediction for a specific coordinate using a neural network parameterized by an implicit function. Then, it predicts a set of modulated weights based on the coordinates and the scale factor through an MLP. These modulated weights are combined with multiple experts to form a modulated filter, which adjusts the predicted values of the queried pixel. The proposed CDCR has two benefits: (1) the dense prediction provides more detailed information to improve prediction accuracy and confidence; and (2) the modulated filter is scale-adaptive and can enhance high-frequency information in the image at a specific resolution. In addition, a scale cumulative ascent (SCA) method is proposed to avoid over-smoothing and enhance the accuracy of predicted images at ultra-high magnifications. The SCA method increases the resolution of the feature map for better dense prediction and eliminates outliers by averaging multiple predictions. As shown in Figure 2, CDCR has a clear advantage with more noticeable details compared to the DR and CR, which is of great help for accuracy-oriented remote sensing scenarios.

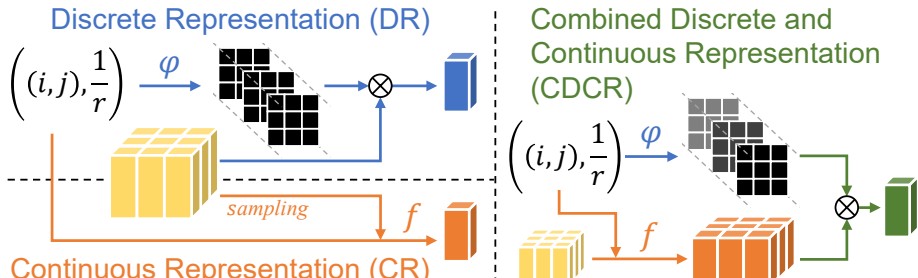

**Figure 1.** Flows of discrete representation (DR), continuous representation (CR), and combined discrete and continuous representation (CDCR). The coordinate of the high-resolution image is represented by $(i, j)$, and the scale factor is represented by $r$. The function $\varphi$ is designed to predict the convolution kernel, while the function $f$ directly maps from the coordinates to signals. In CDCR, the function $\varphi$ predicts a modulated filter that embeds the predicted values into a specified resolution.

The main contributions of this paper can be summarized as follows:

1. A novel approach to image representation, namely CDCR, is proposed, which consists of a CR-based dense prediction and a DR-based resolution-specific refinement. It can be inserted into existing super-resolution frameworks to extend and embed images into any desired resolution.
2. A scale cumulative ascent (SCA) method is introduced to address the underfitting problem at ultra-high magnifications. By aggregating the predictions from various

magnification steps, SCA improves the accuracy and confidence of images reconstructed at ultra-high magnifications.

3. Experiments conducted on publicly available satellite datasets illustrate the generalizability of CDCR. Both qualitative and quantitative evaluations show that CDCR outperforms existing methods across multiple scale factors, making it a more effective method for image representation.

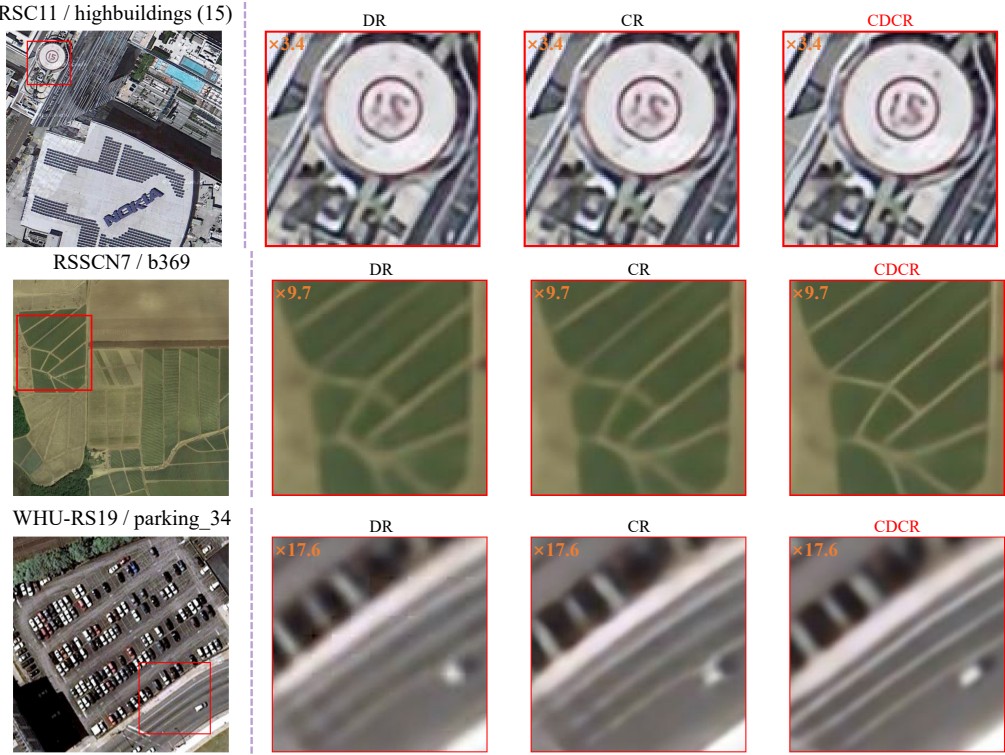

**Figure 2.** Visual comparison between DR, CR, and CDCR at scale factors of 3.4, 9.7, and 17.6, respectively.

## 2. Related Work

### 2.1. Single Image Super-Resolution

Image super-resolution aims to recover high-resolution (HR) images from low-resolution (LR) images, with single image super-resolution (SISR) being a representative topic. Since the publication of the first super-resolution network constructed by convolutional neural networks [3], more and more network structures have been explored, such as residual networks [4,5], recursive networks [6,7], dense connections [8,9], multiple paths [10,11], attention mechanisms [12–14], encoder–decoder networks [15], and Transformer-based networks [16–18]. In addition, other generative models, such as generative adversarial networks (GAN) [19,20], flow-based models [21], and diffusion models [22], have also been applied to SISR tasks.

SISR is a research priority in the field of remote sensing: Lei et al. [23] propose a local-to-global combined network (LGCNet) for learning the multi-level features of salient objects. Jiang et al. [24] introduce a deep distillation recursive network (DDRN), which includes a multi-scale purification unit to compensate for the high-frequency components during information transmission. Lu et al. [25] present a multi-scale residual neural network (MRNN) to compensate for high-frequency information in the generated satellite images. Jiang et al. [26] introduce an edge-enhanced GAN (EEGAN) for recovering sharp edges in images. Wang et al. [27] propose an adaptive multi-scale feature fusion network (AMFFN) to preserve the feature and improve the efficiency of information usage. Several methods also incorporate attention mechanisms: Dong et al. [28] develop a multi-perception atten-

tion network (MPSR), which uses multi-perception learning and multi-level information fusion to optimize the generated images. Zhang et al. [29] present a mixed high-order attention network (MHAN) with an attention module that outperforms channel attention. Ma et al. [30] implement a dense channel attention network (DCAN), in which they design a dense channel attention mechanism to exploit multi-level features. Jia et al. [31] put forward a multi-attention GAN (MA-GAN), which is capable of improving the resolution of images at multiple scale factors. Furthermore, Liu et al. [32] develop a diffusion model with a detail complement mechanism (DMDC), further enhancing the super-resolution effect of small and dense targets in remote sensing images.

## 2.2. Scale-Arbitrary Super-Resolution

The standard SISR process comprises a feature extraction module and a reconstruction module. The core component of the reconstruction module is the upsampling layer, which enhances the resolution of the feature map. Currently, the widely used deconvolutional layer [33] and sub-pixel layer [34] in SISR can only handle fixed integer scale factors, making the model difficult to generalize further. Lim et al. [5] employ multiple upsampling branches for predetermined scale factors, but the method still cannot be applied to arbitrary scale factors.

Meta-SR [1] is the first work aimed at scale-arbitrary super-resolution. In Meta-SR, the meta upscale module utilizes a latent code in LR space to perform a one-to-one mapping of the RGB value in HR space. Because the switch of the latent code is discontinuous, the generated image may contain checkerboard artifacts. Chen et al. [2] propose a local implicit image function (LIIF) to represent the image as a continuous function and introduce a local ensemble to eliminate the checkerboard artifact. However, the use of a shared MLP for each pixel in LIIF neglects the local characteristics of the image, which may result in an overly smooth image. To overcome this problem, Li et al. [35] present an adaptive local image function (A-LIIF), which employs multiple MLPs to model pixel differences and increase the detail in the generated image. Ma et al. [36] introduce the implicit pixel flow (IPF) to convert the original blurry implicit neural representation into a sharp one, resolving the problem of overly smooth images generated by LIIF.

In addition to building upsampling modules as in Meta-SR and LIIF, some approaches introduce scale information into the feature extraction module to create scale-aware feature extraction modules. Such modules include the scale-aware dynamic convolutional layer for feature extraction [37], the scale attention module that adaptively rescales the convolution filters [38], and the scale-aware feature adaption blocks based on conditional convolution [39], among others. These scale-aware modules align with the target resolution, improving the learning ability of the network for arbitrary scale factors. To sum up, Meta-SR and its subsequent works significantly advance the field of scale-arbitrary super-resolution by overcoming the limitations of fixed scale factors and improving the quality of the generated images.

The studies in remote sensing have not fully explored the concept of scale-arbitrary super-resolution. For instance, Fang et al. [40] propose an arbitrary upscale module based on Meta-SR and add an edge reinforcement module in post-processing stage to enhance the high-frequency information of the generated images. In addition, He et al. [41] present a video satellite image framework that enhances spatial resolution by subpixel convolution and bicubic-based adjustment. To conclude, there remains a requirement for a general approach to tackle the challenge of scale-arbitrary super-resolution for satellite images.

## 2.3. Image Rescaling

Compared to super-resolution models that primarily focus on image upscaling, image rescaling (IR) integrates both image downscaling and upscaling to achieve more precise preservation of details. Therefore, the upscaling component of IR can also be used for super-resolution reconstruction of images at arbitrary resolutions. Xiao et al. [42] developed an invertible rescaling net (IRN) with a deliberately designed framework but limit it to a

fixed integer scale factor. In contrast, Pan et al. [43] proposed a bidirectional arbitrary image rescaling network (BAIRNet) that unifies image downscaling and upscaling as a single learning process. Later, Pan et al. [44] introduced a simple and effective invertible arbitrary rescaling network (IARN) that achieves arbitrary image rescaling with better performance than BAIRNet. In the field of remote sensing, Zou et al. [45] proposed a rescaling-assisted image super-resolution method (RASR) to better restore lost information in medium-low resolution remote sensing images.

As a newly emerged field, IR requires modeling the process of downscaling images and supporting image magnification at non-integer scale factors. However, most IR methods mainly focus on exploring continuous representations of images at lower scale factors (e.g., less than $4\times$) and neglect the potential underfitting problem that may arise at higher scale factors.

## 3. Methods

In this section, we first formally define the discrete and continuous representation in scale-arbitrary super-resolution. Subsequently, we introduce a novel approach that leverages a combination of the discrete and continuous representation, and provide an in-depth explanation of its individual components. Lastly, we examine the underfitting problem at ultra-high magnifications and suggest a scale cumulative ascent method as a practical approach to mitigate the problem.

### 3.1. Discrete Representation (DR) and Continuous Representation (CR)

Scale-arbitrary super-resolution aims to enlarge a low-resolution image by a scale factor of $r$. Suppose a low-resolution image $X \in \mathbb{R}^{h \times w \times c}$ can be encoded into 2D features $F \in \mathbb{R}^{h \times w \times d}$. In that case, a neural network-parameterized decoder $\phi$ can be used to convert the features $F$ into the corresponding high-resolution image $Z \in \mathbb{R}^{\lfloor rh \rfloor \times \lfloor rw \rfloor \times c}$, where $h$ and $w$ represent the height and width of the image or feature, while $d$ and $c$ represent the depth of the features and the number of channels in the image, respectively. Therefore,

$$Z = \phi(F, r). \tag{1}$$

The mapping function $\phi : F \mapsto Z$ can be either a discrete mapping based on DR (denoted as $\phi_D$) or a continuous mapping based on CR (denoted as $\phi_C$). The difference between the two is noted below.

#### 3.1.1. Discrete Representation (DR)

DR is designed to perform a discrete mapping from low-resolution (LR) space to high-resolution (HR) space. Given a scale factor of $r$, the resolution of the LR and HR space can be determined. Next, DR needs to match each coordinate $x = (i, j)$ in the HR space to its corresponding coordinate $x' = (i', j')$ in the LR space. Let $\mathcal{T}$ be the coordinate mapping function, then $\mathcal{T}(x, r) = x'$. When the value of $r$ is fixed, the function $\mathcal{T}(x, r) = x'$ can be simplified as $\mathcal{T}(x) = x'$ to emphasize the mapping relationship between the original input $x$ and its transformed output $x'$. The RGB value of the HR image at coordinate $x$ can be predicted based on the feature $F[\mathcal{T}(x)]$ at coordinate $\mathcal{T}(x)$. The discrete kernel is defined as $W(x)$. As a result, Equation (1) can be updated to

$$W(x) = \varphi(x - \mathcal{T}(x)), \tag{2}$$
$$Z(x) = \phi_D(F[\mathcal{T}(x)], W(x)) = F[\mathcal{T}(x)] \cdot W(x), \tag{3}$$

where $W(x)$ is a set of dynamic filters predicted based on the coordinate offset $\Delta x = x - T(x)$. $W(x)$ serves two purposes: first, it corrects the coordinate offset $\Delta x$ caused by non-integer $r$ in the coordinate matching; second, it reduces the number of channels $d$ in the feature map to fit the number of channels $c$ in the predicted image.

### 3.1.2. Continuous Representation (CR)

CR aims to predict a continuous mapping function $\phi_C$. Typically, $\phi_C$ is an implicit neural representation parameterized by a neural network that represents an image as a function $f : \mathcal{X} \mapsto \mathcal{S}$ mapped from the coordinate domain $\mathcal{X}$ to the signal domain $\mathcal{S}$, i.e.,

$$s = f(v, x), \tag{4}$$

where $v$ is the code vector, $x = (i, j) \in \mathcal{X}$ represents the 2D coordinates of the HR image, and $s = (s_r, s_g, s_b) \in \mathcal{S}$ refers to the RGB value of the HR image at coordinate $x$. Firstly, assume that the 2D coordinates of the feature extracted from the image are uniformly distributed, so the coordinates $x'$ in LR space and $x$ in HR space can be normalized to a range of $[-A, A]$ ($A$ is a predefined positive value) to obtain $\tilde{x}'$ and $\tilde{x}$, respectively. Due to the continuity of the coordinates, the implicit neural representation naturally suits the continuous representation of images. Then, according to the nearest neighbor function $\mathcal{U}$, the coordinate $\tilde{x}'$ closest to $\tilde{x}$ is obtained by $\mathcal{U}(\tilde{x}, r) = \tilde{x}'$, where $r$ represents the scale factor. Similar to Section 3.1.1, $\mathcal{U}(\tilde{x}, r) = \tilde{x}'$ can be simplified as $\mathcal{U}(\tilde{x}) = \tilde{x}'$. Finally, the latent code required for the implicit neural representation comes from the feature $F[\mathcal{U}(\tilde{x})]$ at coordinate $\mathcal{U}(\tilde{x})$. As a result, Equation (4) can be revised as

$$Z(\tilde{x}) = \phi_C(F[\mathcal{U}(\tilde{x})], \tilde{x} - U(\tilde{x})). \tag{5}$$

In summary, DR is a two-stage process that predicts the discrete kernel $W(x)$ at coordinate $x$ in a high-resolution image, while CR is a one-stage process that directly maps the normalized continuous coordinate $\tilde{x}'$ to RGB values $Z(\tilde{x}')$. In terms of super-resolution performance, DR can optimize the performance at arbitrary resolutions through resolution-specific kernels and is more effective at smaller scales, while CR predicts a more general representation, resulting in better performance at larger scales due to its strong generalization. Experiments conducted by Chen et al. [2] confirm this view.

### 3.2. Combined Discrete and Continuous Representation (CDCR)

Based on the aforementioned studies, this paper proposes a method that combines discrete and continuous representation, referred to as CDCR. In CDCR, CR guarantees the accuracy of high-magnification predictions, while DR fine-tunes the generated image and strengthens high-frequency information at the desired resolution.

### 3.2.1. CR-Based Dense Prediction

Figure 3 depicts the proposed CDCR method. The coordinate $\tilde{x}$ in HR space and the coordinate $\tilde{x}'$ in LR space are normalized to the interval $[-A, A]$. In contrast to the standard CR, for each coordinate $\tilde{x}$ in HR space, we perform a dense prediction of the RGB values for a set of $3 \times 3$ pixels centered at $\tilde{x}$. To this end, we refer to the work of Chen et al. [2] and expand the latent code and coordinate information specified in Equation (5). On the one hand, we concatenate the adjacent latent code, i.e., expand the number of feature channels from $d$ to $9d$:

$$\tilde{F}_{mn} = \text{Concat}\left(\left\{F_{m+p, n+q}\right\}_{p,q \in \{-1,0,1\}}\right); \tag{6}$$

On the other hand, we obtain the position coordinates $\mathcal{U}_t(\tilde{x})$ of the four closest latent codes to $\tilde{x}$ through the nearest neighbor function $\mathcal{U}_t$ that is oriented in different directions. Note that $t = 00, 01, 10,$ and $11$ represent four directions of the top left, top right, bottom left, and bottom right of the queried pixel, respectively. In Figure 3, these latent codes are identified as $z_{00}^*, z_{01}^*, z_{10}^*,$ and $z_{11}^*$. We incorporate the coordinate offset $\tilde{x} - \mathcal{U}_t(\tilde{x})$ and the shape $\tilde{c}$ of the queried pixel at coordinates $\tilde{x}$ into the 2D coordinate information $\zeta_t(\tilde{x})$, that is,

$$\zeta_t(\tilde{x}) = \text{Concat}(\tilde{x} - \mathcal{U}_t(\tilde{x}), \tilde{c}), \tag{7}$$

where $\tilde{c} = (A/hr, A/wr)$ depends on the scale factor $r$, indicating the height and width of the queried pixel. As a result, the CR-based dense prediction can be represented as

$$Z_t^*(\tilde{x}) = \phi_+\big(\tilde{F}[\mathcal{U}_t(\tilde{x})], \zeta_t(\tilde{x})\big) \tag{8}$$

$$M(\tilde{x}) = \sum_{t \in \{00,01,10,11\}} \frac{S_{t'}}{S} \cdot Z_t^*(\tilde{x}) \tag{9}$$

where $\phi_+$ is an enhanced dense prediction that yields a greater number of channels. The definition of $Z_t^*$ comes from Equation (5). The factor $S_{t'}$ serves as a weight, with $t'$ being diagonal to $t$ (i.e., '10' to '10', '00' to '11') and $S = \sum_t S_t$. The output $M(\tilde{x})$ in Equation (9) represents the RGB predictions of $3 \times 3$ pixels centered at $\tilde{x}$. The CR-based dense prediction provides the foundation for the DR-based resolution-specific refinement.

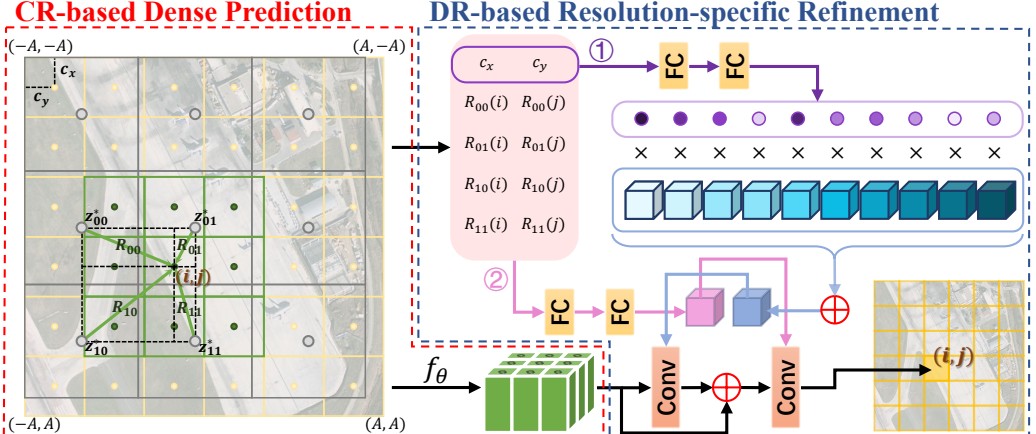

**Figure 3.** The structure of CDCR. CDCR consists of two parts: a CR-based dense prediction and a DR-based resolution-specific refinement. In the first part, we derive a set of $3 \times 3$ predictions centered at coordinate $(i, j)$ through the dense prediction $f_\theta$. In the second part, we adjust the predictions for the given resolution by means of two branches: Branch 1 (designated by symbol ①) pre-modulates the features based on scale information, while Branch 2 (designated by symbol ②) further embeds the predicted pixels into the specified resolution by leveraging the coordinate offsets and modulated features.

### 3.2.2. DR-Based Resolution-Specific Refinement

CR may result in overly smoothed predictions. To enhance the high-frequency information in the images, we perform DR-based resolution-specific refinement on the dense predicted features obtained at each coordinate $\tilde{x}$ in HR space.

As shown in Figure 3, the proposed DR-based resolution-specific refinement contains two branches: Branch 1 aims at resolution awareness and feature modulation, while Branch 2 primarily enhances high-frequency information in the generated images. The work of Wang et al. [39] demonstrates that features learned from images are different for various target resolutions. In other words, the features required by the network vary for different resolutions. Hence, we pre-modulate the dense predicted features based on scale information in Branch 1: Initially, the shape of the queried pixel is fed into a modulator composed of two fully connected layers to generate modulated weights $p_i$. Subsequently, these resolution-based modulated weights $p_i$ and experts $P_i$ are combined into a scale-modulated filter to pre-modulate the dense predicted features $M$, i.e.,

$$M_e = M + M * \left( \frac{1}{k} \sum_{i=1}^{k} p_i \cdot P_i \right), \tag{10}$$

where $M_e$ stands for the dense features with scale awareness. The experts contain $k$ convolution kernels that are trained to recognize various resolutions. The modulated feature $M_e$ provides better discrimination compared to $M$ and forms the foundation for learning high-frequency information at a specific resolution.

In Branch 2, we use DR to predict the RGB value of the queried pixel at coordinate $\tilde{x}$: Firstly, we establish a set of coordinate offsets $\varpi$ based on $\tilde{x}$:

$$\varpi = \text{Concat}\Big(\{R_t\}_{t\in\{00,01,10,11\}}, \tilde{c}\Big), \tag{11}$$

where $R_t = \tilde{x} - \mathcal{U}_t(\tilde{x})$ represents the distance between the queried pixel and the latent code $z_t^*$. $\tilde{c}$ denotes the height and width of the queried pixel. Then, we directly predict the RGB value of each pixel from the coordinate offset $\varpi$ and the modulated feature $M_e$. According to Equation (3), the DR-based resolution-specific refinement can be simplified to

$$Z(\tilde{x}) = M_e[(\tilde{x})] \cdot \varphi(\varpi). \tag{12}$$

In short, the prediction of the modulated filter is carried out in Branch 1 based on scale information, while the prediction of high-frequency information is conducted in Branch 2 through DR. The network effectively captures the residual high-frequency information between the smoothed prediction and the ground truth, reducing over-smoothing and decreasing the learning difficulty.

## 4. Discussions: The Underfitting Problem at Ultra-High Magnifications

The performance of existing models at ultra-high magnifications (e.g., $r \geq 8$) remains inadequate due to the persistent underfitting problem. Regrettably, there is a scarcity of studies that address this problem. Our investigation reveals that the main causes of underfitting at ultra-high magnifications are: (1) the model tries to fit low magnifications (in-distribution) during training, resulting in neglect of the generalizability of high magnifications (out-of-distribution); (2) the resolution of the feature (i.e., latent code) is significantly lower compared to the predicted image, i.e., the feature coordinates are too sparse, leading to excessive utilization of each feature vector.

The majority of scale-arbitrary super-resolution methods [1,2,35,37] set $1 < r \leq 4$ as in-distribution and $r \geq 4$ as out-of-distribution, and this setting is followed in this paper. We define the probability of a set of $3 \times 3$ pixels in HR space crossing a matrix array connected by latent codes as $\psi$. Figure 4a illustrates a qualitative comparison between in-distribution and out-of-distribution to demonstrate the sparsity of latent code coordinates. Figure 4b provides a quantitative computation of the sparsity index $\psi$. The sparsity of the latent code increases as $\psi$ decreases. As depicted in Figure 4b, $\psi$ displays substantial differences between in-distribution and out-of-distribution, e.g., $\psi = 0.94$ for $r = 4$, while $\psi = 0.27$ for $r = 20$. To mitigate the rapid decrease in $\psi$, we introduce a scale cumulative ascent (SCA) method to enhance the prediction ability of the proposed CDCR in out-of-distribution scenarios.

SCA achieves an ultra-high magnification of images through a stepwise increase in resolution. As illustrated in Figure 5, the initial CDCR module enhances the density of the latent codes, providing more comprehensive feature information for subsequent CDCR modules. This approach significantly improves the accuracy of high-resolution image reconstruction during the upsampling process. In this paper, we establish multiple serial stages of magnification with multiple scale factors $r_1, r_2, \ldots$, where $r = \prod r_i$. The SCA method divides the scale factor $r$ into the following format:

$$r = \begin{cases} r_1 r_2 = (r_1 - \delta) \cdot \left(r_2 + \frac{r_2\delta}{r_1-\delta}\right), & r_m < r \leq r_M \\ r_1 r_2 r_3 = (r_1 - \gamma) \cdot r_2 \cdot \left(r_3 + \frac{r_3\gamma}{r_1-\gamma}\right), & r \geq r_M \end{cases} \tag{13}$$

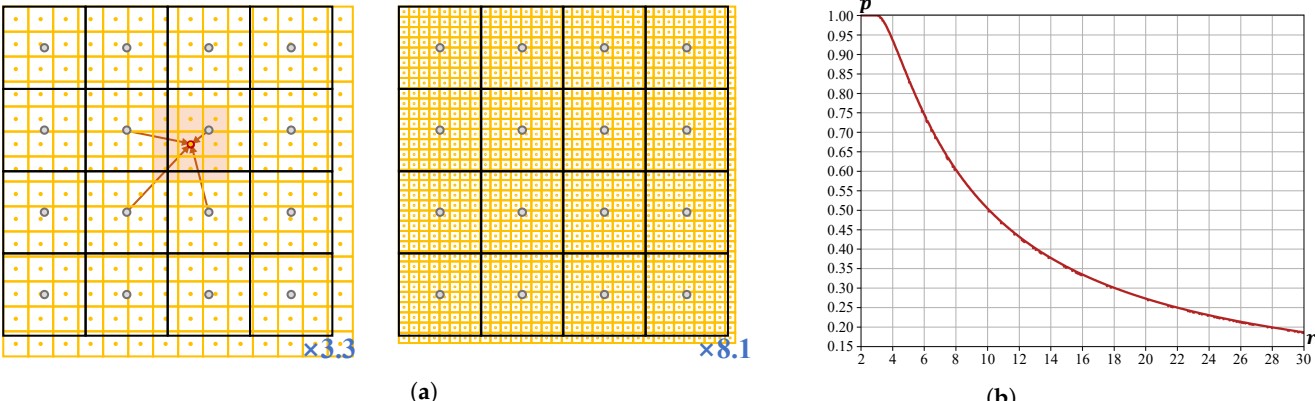

(a)　　　　　　　　　　　　　　　　　　　　　　　　　　　　　(b)

**Figure 4.** The sparsity of the latent code at different scale factors. Figure (**a**) shows the density of the LR coordinates compared to the HR spatial coordinates at scale factors of 3.3 and 8.1. The sparsity index $\psi$ is calculated in Figure (**b**) to demonstrate the density of the latent code at different resolutions.

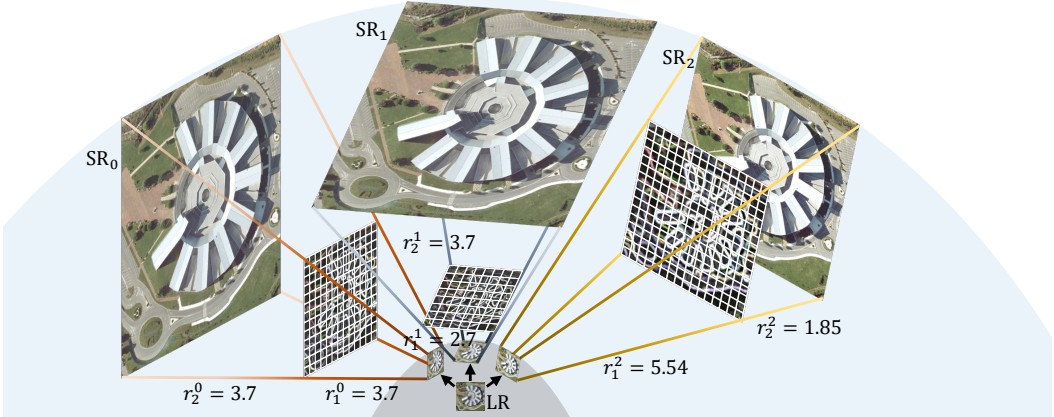

**Figure 5.** The proposed scale cumulative ascent (SCA) method operates at a scale factor of 9.7. The figure depicts that LR in the low-resolution manifold space is enlarged through three pathways, resulting in $SR_0$, $SR_1$, and $SR_2$ in the high-resolution manifold space. Here, $r_i^j$ denotes the $i$-th scale factor of the $j$-th pathway. The final prediction is obtained by averaging these SR images. On the one hand, SCA improves the effectiveness of dense prediction and increases accuracy in out-of-distribution scenarios by augmenting the density of the latent code. On the other hand, as a single LR image may correspond to multiple HR images, utilizing SCA to average multiple predictions enhances the robustness of the generated HR images.

In Equation (13), the scale factor $r$ is decomposed into multiple $r_i$. If $r_m < r \leq r_M$, SCA performs a two-step magnification, while if $r \geq r_M$, it performs a three-step magnification. $\delta = \{\delta_1, \delta_2, ...\}$ and $\gamma = \{\gamma_1, \gamma_2, ...\}$ are predetermined. The SCA method has three advantages: (1) it makes full use of the model's strong ability to fit data within in-distribution; (2) it effectively increases the density of the latent code, leading to improved results for dense predictions; and (3) it reduces the risk of outliers in out-of-distribution scenarios by averaging the predictions obtained from different magnification sequences.

## 5. Experiments

In this section, we first provide an overview of the experimental datasets and training parameters. Then, we compare our proposed CDCR method with current state-of-the-art scale-arbitrary super-resolution methods. After that, we evaluate the impact of the CR-based dense prediction, DR-based resolution-specific refinement, encoder, and SCA on prediction accuracy. Finally, we analyze the complexity of the methods.

### 5.1. Datasets and Metrics

We employ the AID dataset [46] for training in this study. AID is a large-scale aerial image dataset consisting of 10,000 images with a resolution of $600 \times 600$ pixels, covering 30 classes of scenes. The dataset includes images collected by various remote sensing sensors, each with a spatial resolution of 0.5–0.8 meters, covering diverse seasons and time-frames.

The test datasets include RSC11 [47], RSSCN7 [48], and WHU-RS19 [49]: (1) The RSC11 dataset [47] comprises a total of 1232 high-resolution remote sensing images, which cover 11 distinct scene categories and multiple cities in the United States. Each image has a size of $512 \times 512$ pixels and a spatial resolution of 0.2 meters. (2) The RSSCN7 dataset [48] is comprised of 2800 remote sensing images, which are classified into seven distinct scene categories. Each image has a size of $400 \times 400$ pixels. Within each category, the images are collected using four different scale variations and from multiple imaging angles. (3) The WHU-RS19 dataset [49] consists of 1005 remote sensing images covering 19 land use categories, each with a size of $600 \times 600$ pixels. Images in each category are acquired from diverse geographic locations, exhibiting variations in scale (with a maximum spatial resolution of 0.5 m) and illumination conditions.

It is noteworthy that the aforementioned datasets are acquired from Google Earth (Google Inc.). These images may have undergone several pre-processing techniques, such as geometric correction, noise reduction, and color balancing, and have been subjected to image compression (e.g., conversion from high bit depth to 8-bit depth) to optimize storage and transmission efficiency.

Given the high demand for image accuracy in remote sensing imagery, we utilize peak signal-to-noise ratio (PSNR) in decibels (dB) as a measure of image accuracy. A higher PSNR value indicates greater image accuracy.

### 5.2. Implementation Details

During the training phase, the scale factor is established within a range of $\times 1$ to $\times 4$ (in-distribution). During the testing phase, the scale factor extends to $\times 6 \sim \times 20$ (out-of-distribution) beyond the training range. Each low-resolution (LR) image patch is of size $48 \times 48$ and is derived through bicubic downsampling of the corresponding high-resolution (HR) image. The value of variable $A$ is set to 1. A random scale factor, denoted as $r_b$, is sampled for each batch from a uniform distribution ranging from $\times 1$ to $\times 4$, resulting in HR image patches with a size of $\lfloor 48r_b \rfloor \times \lfloor 48r_b \rfloor$ pixels. Subsequently, the HR images are transformed into pairs of coordinates and RGB values, and a random sample of $48^2$ (equal to 2304) is selected. The loss function adopted is the $\ell_1$ loss, and the optimizer utilized is ADAM with an initial learning rate $\eta = 0.0001$. The training period is set for 1000 epochs, with a reduction in the learning rate by half every 200 epochs. The encoder employed is the EDSR [5] model with 16 residual blocks. For details on the SCA configuration, please see Section 5.4.4. The code will be made publicly available at https://github.com/Suanmd/CDCR/.

### 5.3. Comparing Methods

In this part, we compare CDCR with the dominant scale-arbitrary super-resolution methods. The compared methods are:

1. Bicubic: The baseline method that works for any scale factors.
2. Meta-SR [1]: The representative method for DR.
3. LIIF [2]: The representative method for CR.
4. ArbSR [39]: A scale awareness method based on DR. The scale-aware upsampling layer of the method is extracted for comparison.
5. A-LIIF [35]: An adaptive local method based on CR. It models pixel differences through multiple MLPs to eliminate possible artifacts in LIIF.

6.  CDCR (ours): The method proposed in this paper combines both DR and CR. It involves a CR-based dense prediction and a DR-based resolution-specific refinement but does not include SCA.

The above methods are all implemented with the same settings described in Section 5.2 to ensure fairness in the comparison. The quantitative results are shown in Table 1.

**Table 1.** Quantitative comparison of various methods across multiple scale factors using PSNR(dB). The best values for each scale factor across different datasets are highlighted in **bold**.

| Dataset | Method | In-Distribution | | | | Out-Of-Distribution | | | |
|---|---|---|---|---|---|---|---|---|---|
| | | ×2 | ×3 | ×4 | ×6 | ×8 | ×12 | ×16 | ×20 |
| RSC11 [47] | Bicubic | 30.33 | 27.31 | 25.75 | 24.16 | 23.26 | 22.19 | 21.51 | 21.05 |
| | Meta-SR [1] | 33.23 | 29.52 | 27.57 | 25.46 | 24.30 | 22.97 | 22.19 | 21.63 |
| | LIIF [2] | 33.21 | 29.52 | 27.58 | 25.50 | 24.35 | 23.01 | 22.23 | 21.66 |
| | ArbSR [39] | 33.20 | 29.50 | 27.55 | 25.44 | 24.29 | 22.96 | 22.18 | 21.62 |
| | A-LIIF [35] | 33.23 | 29.52 | 27.58 | 25.49 | 24.36 | 23.02 | 22.23 | 21.67 |
| | CDCR (ours) | **33.26** | **29.58** | **27.63** | **25.54** | **24.39** | **23.05** | **22.25** | **21.68** |
| RSSCN7 [48] | Bicubic | 29.96 | 27.47 | 26.16 | 24.75 | 23.92 | 22.95 | 22.35 | 21.94 |
| | Meta-SR [1] | 32.14 | 29.01 | 27.42 | 25.68 | 24.69 | 23.54 | 22.84 | 22.38 |
| | LIIF [2] | 32.15 | 29.02 | 27.44 | 25.71 | 24.73 | 23.58 | 22.88 | 22.42 |
| | ArbSR [39] | 32.11 | 29.00 | 27.40 | 25.66 | 24.68 | 23.53 | 22.84 | 22.38 |
| | A-LIIF [35] | 32.14 | 29.01 | 27.44 | 25.71 | 24.73 | 23.58 | 22.87 | 22.42 |
| | CDCR (ours) | **32.18** | **29.04** | **27.46** | **25.74** | **24.75** | **23.59** | **22.88** | **22.43** |
| WHU-RS19 [49] | Bicubic | 33.47 | 29.87 | 27.92 | 25.84 | 24.69 | 23.37 | 22.61 | 22.06 |
| | Meta-SR [1] | 36.56 | 32.24 | 29.91 | 27.30 | 25.86 | 24.24 | 23.33 | 22.70 |
| | LIIF [2] | 36.55 | 32.24 | 29.94 | 27.35 | 25.93 | 24.30 | 23.39 | 22.75 |
| | ArbSR [39] | 36.53 | 32.21 | 29.89 | 27.28 | 25.85 | 24.23 | 23.33 | 22.70 |
| | A-LIIF [35] | 36.55 | 32.24 | 29.93 | 27.35 | 25.92 | 24.30 | 23.39 | 22.75 |
| | CDCR (ours) | **36.62** | **32.28** | **29.97** | **27.39** | **25.95** | **24.31** | **23.40** | **22.77** |

Table 1 demonstrates the superiority of CDCR. CR performs optimally at high magnification levels, while DR may exhibit better results at low magnifications. By integrating these two characteristics, CDCR enhances the prediction results for most scale factors. Compared to the baseline, the improvement decreases as the scale factor increases.

The following pages present a qualitative comparison of the methods. We evaluate scale factors of ×4/×8 (refer to Figure 6) and ×12/×20 (refer to Figure 7) on the test datasets. The results demonstrate that CDCR offers significant improvement in some specific scenes, such as the edges of vehicles and lines of lanes.

### 5.4. Ablation Study

In this section, we examine the importance of the individual components of CDCR and demonstrate the improved results brought about by the integration of SCA.

#### 5.4.1. Analysis of CR-Based Dense Prediction

The CR-based dense prediction integrates more information to enhance its ability to predict uncertain pixels. In Equation (9), $M(\tilde{x})$ denotes the predicted values for a set of 9 pixels centered at $\tilde{x}$. In practice, the density of the prediction needs to be considered. To this end, we conduct experiments in three groups: the first group involves no dense prediction, i.e., predicting the RGB value for the pixel at $\tilde{x}$ (denoted as CDCR-c1); the second group involves semi-dense prediction, i.e., predicting the RGB values of a set of 4 pixels centered at $\tilde{x}$ (denoted as CDCR-c4); and the third group involves dense prediction, i.e., predicting the RGB values of a set of 9 pixels centered at $\tilde{x}$ (denoted as CDCR-c9 and set as the default). The results are presented in Table 2.

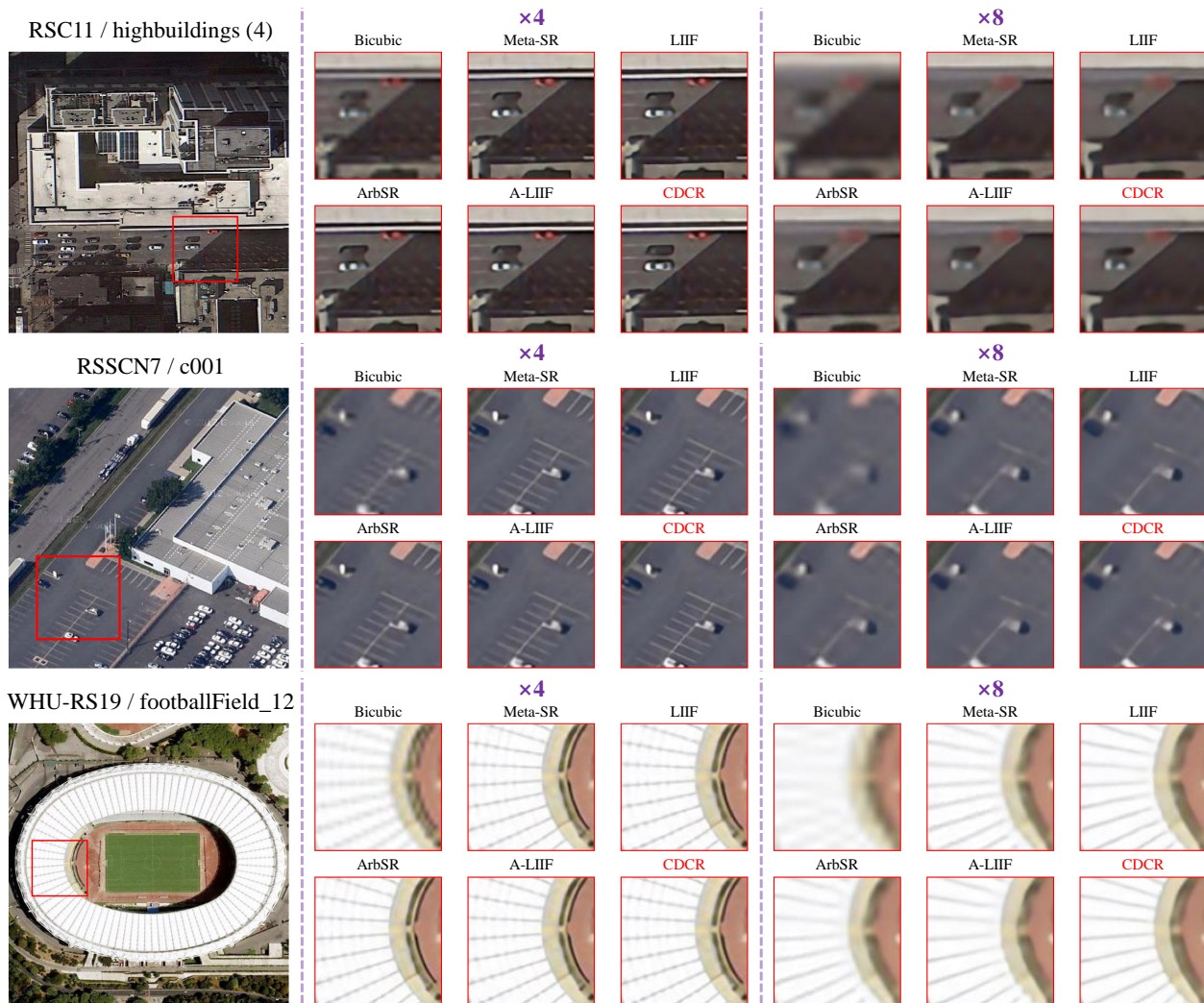

**Figure 6.** Visual comparison of the different methods at scale factors of 4 and 8.

On the one hand, the enhancement from CDCR-c1 to CDCR-c4 is substantial, indicating the effectiveness of dense prediction. On the other hand, the enhancement observed in CDCR-c9 compared to CDCR-c4 appears to be minimal, implying a diminishing return of the dense prediction.

**Table 2.** Ablation experiment on dense prediction. -c1 represents no dense prediction, -c4 represents semi-dense prediction, and -c9 represents dense prediction. The optimal PSNR values for each scale factor across different datasets are emphasized in **bold**.

| | | In-Distribution | | | Out-Of-Distribution | | | | |
|---|---|---|---|---|---|---|---|---|---|
| **Dataset** | **Method** | $\times 2$ | $\times 3$ | $\times 4$ | $\times 6$ | $\times 8$ | $\times 12$ | $\times 16$ | $\times 20$ |
| RSC11 [47] | CDCR-c1 | 33.22 | 29.55 | 27.60 | 25.52 | 24.37 | 23.03 | 22.24 | 21.67 |
| | CDCR-c4 | 33.26 | 29.58 | 27.63 | 25.54 | 24.39 | 23.04 | 22.24 | 21.67 |
| | CDCR-c9 | **33.26** | **29.58** | **27.63** | **25.54** | **24.39** | **23.05** | **22.25** | **21.68** |
| RSSCN7 [48] | CDCR-c1 | 32.15 | 29.03 | 27.45 | 25.72 | 24.73 | 23.58 | 22.88 | 22.42 |
| | CDCR-c4 | 32.18 | 29.04 | 27.46 | 25.73 | 24.75 | 23.59 | 22.88 | 22.42 |
| | CDCR-c9 | **32.18** | **29.04** | **27.46** | **25.74** | **24.75** | **23.59** | **22.88** | **22.43** |
| WHU-RS19 [49] | CDCR-c1 | 36.56 | 32.26 | 29.94 | 27.36 | 25.93 | 24.30 | 23.39 | 22.76 |
| | CDCR-c4 | 36.60 | 32.28 | 29.97 | 27.38 | 25.95 | 24.31 | 23.40 | 22.77 |
| | CDCR-c9 | **36.62** | **32.28** | **29.97** | **27.39** | **25.95** | **24.31** | **23.40** | **22.77** |

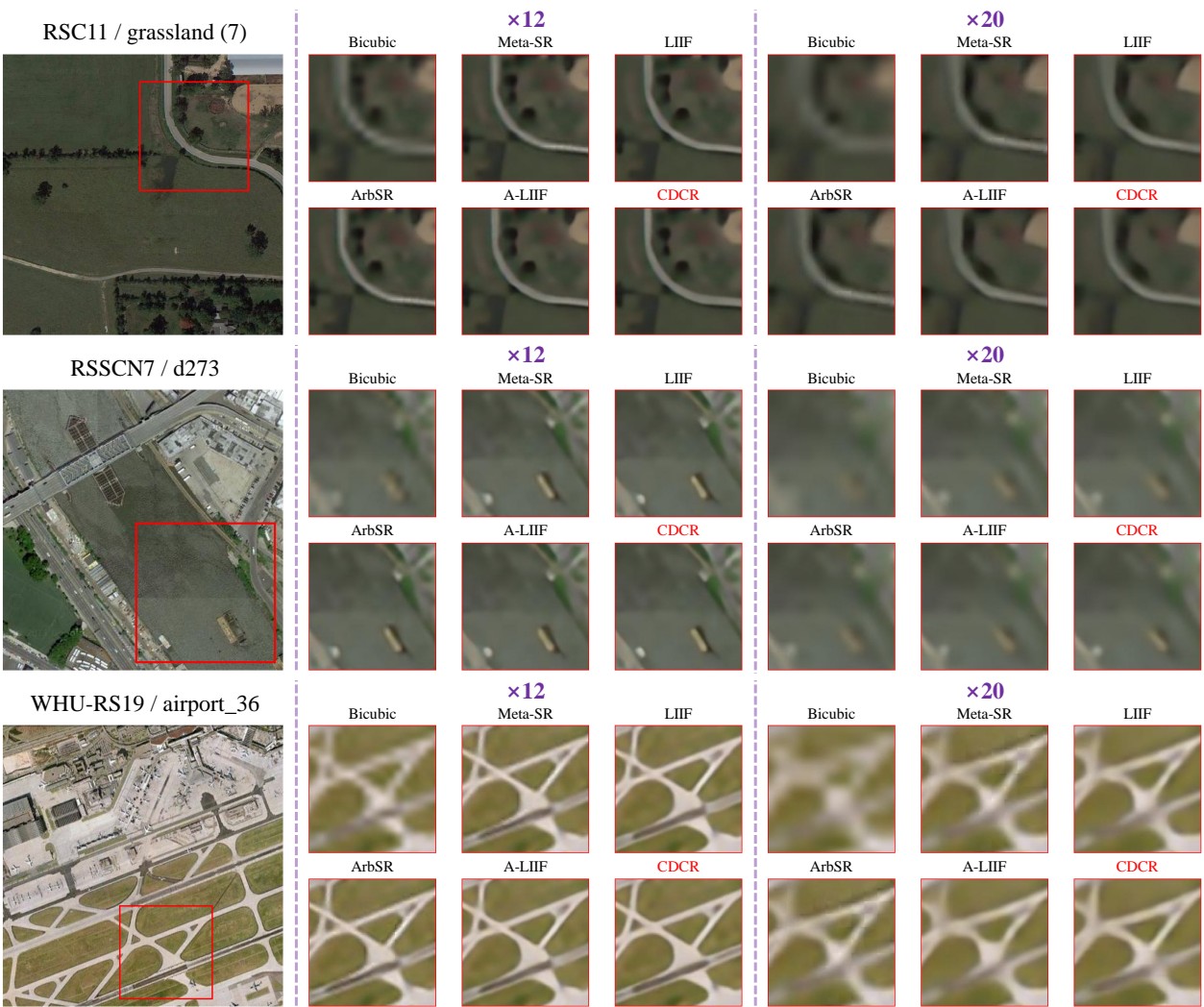

**Figure 7.** Visual comparison of the different methods at scale factors of 12 and 20.

### 5.4.2. Analysis of DR-Based Resolution-Specific Refinement

As shown in Figure 3, there are two branches in the DR-based resolution-specific refinement process. Branch 2 is essential to DR, while Branch 1 is introduced to pre-modulate the features in a scale-aware manner. The pre-modulation in Branch 1 integrates $k$ experts, each with a varying focus on different scales. To assess the impact of the resolution-specific pre-modulation, we conduct three groups of experiments: the first experiment omits pre-modulation (denoted as CDCR-k0); the second experiment utilizes $k = 3$ (denoted as CDCR-k3); the third experiment utilizes $k = 10$ (denoted as CDCR-k10 and set as the default). The quantitative results are presented in Table 3.

Table 3 highlights the requirement for resolution-specific modulation. In other words, the combination of CR and DR through the utilization of pre-modulation is essential.

**Table 3.** Ablation experiment on pre-modulation in resolution-specific refinement. -k0 indicates the absence of pre-modulation, while -k3 and -k10 indicate the integration of 3 and 10 experts for pre-modulation. The optimal PSNR values for each scale factor across different datasets are emphasized in **bold**.

| Dataset | Method | In-Distribution | | | Out-Of-Distribution | | | | |
| --- | --- | --- | --- | --- | --- | --- | --- | --- | --- |
| | | ×2 | ×3 | ×4 | ×6 | ×8 | ×12 | ×16 | ×20 |
| RSC11 [47] | CDCR-k0 | 33.23 | 29.52 | 27.60 | 25.51 | 24.36 | 23.02 | 22.23 | 21.65 |
| | CDCR-k3 | **33.27** | 29.57 | 27.63 | 25.54 | 24.39 | 23.03 | 22.24 | 21.67 |
| | CDCR-k10 | 33.26 | **29.58** | **27.63** | **25.54** | **24.39** | **23.05** | **22.25** | **21.68** |
| RSSCN7 [48] | CDCR-k0 | 32.15 | 29.02 | 27.45 | 25.72 | 24.73 | 23.58 | 22.87 | 22.41 |
| | CDCR-k3 | **32.19** | **29.05** | **27.46** | 25.73 | 24.75 | 23.59 | 22.88 | 22.43 |
| | CDCR-k10 | 32.18 | 29.04 | 27.46 | **25.74** | **24.75** | **23.59** | **22.88** | **22.43** |
| WHU-RS19 [49] | CDCR-k0 | 36.53 | 32.24 | 29.93 | 27.36 | 25.92 | 24.30 | 23.38 | 22.75 |
| | CDCR-k3 | **36.62** | **32.29** | 29.97 | 27.38 | 25.94 | 24.31 | 23.40 | 22.76 |
| | CDCR-k10 | 36.62 | 32.28 | **29.97** | **27.39** | **25.95** | **24.31** | **23.40** | **22.77** |

### 5.4.3. Analysis of Encoder

A stronger encoder leads to more powerful latent codes, thus enhancing the effect of the predicted results. In this research, we select three encoders, namely EDSR [5], RDN [9], and RCAN [12]. The number of channels in the feature maps is set to 64. The configuration of the EDSR model consists of 16 residual blocks, as described in Section 5.2. The RDN model is structured with 16 residual dense blocks, each composed of 8 convolutional layers. The RCAN model comprises 10 residual groups, each consisting of 20 residual channel attention blocks. The performance of Meta-SR, LIIF, and CDCR on the RSC11 dataset is assessed in Table 4.

Table 4 clearly shows the significant influence of the encoder on the outcome. Despite its high complexity, the RCAN model performs best in our experiment. Moreover, Table 4 confirms the strong generalizability of our proposed method.

**Table 4.** Ablation experiment on encoder. The encoders employed in this experiment are sourced from three SISR frameworks: EDSR, RDN, and RCAN. The optimal PSNR values for each scale factor are shown in **bold**.

| Dataset | Method | In-Distribution | | | Out-Of-Distribution | | | | |
| --- | --- | --- | --- | --- | --- | --- | --- | --- | --- |
| | | ×2 | ×3 | ×4 | ×6 | ×8 | ×12 | ×16 | ×20 |
| Meta-SR [1] | EDSR [5] | 33.23 | 29.52 | 27.57 | 25.46 | 24.30 | 22.97 | 22.19 | 21.63 |
| | RDN [9] | 33.44 | 29.71 | 27.77 | 25.63 | 24.47 | 23.09 | 22.27 | 21.70 |
| | RCAN [12] | **33.51** | **29.77** | **27.79** | **25.66** | **24.48** | **23.10** | **22.29** | **21.72** |
| LIIF [2] | EDSR [5] | 33.21 | 29.52 | 27.58 | 25.50 | 24.35 | 23.01 | 22.23 | 21.66 |
| | RDN [9] | 33.39 | 29.68 | 27.76 | 25.65 | 24.49 | 23.13 | 22.32 | 21.74 |
| | RCAN [12] | **33.47** | **29.75** | **27.82** | **25.72** | **24.56** | **23.17** | **22.35** | **21.77** |
| CDCR (ours) | EDSR [5] | 33.26 | 29.58 | 27.63 | 25.54 | 24.39 | 23.05 | 22.25 | 21.68 |
| | RDN [9] | 33.41 | 29.71 | 27.77 | 25.67 | 24.51 | 23.14 | 22.33 | 21.76 |
| | RCAN [12] | **33.50** | **29.79** | **27.83** | **25.72** | **24.57** | **23.18** | **22.35** | **21.78** |

### 5.4.4. Analysis of SCA

The paper states that the purpose of SCA is to enhance the dense prediction during the inference phase. SCA allows the combination of multiple magnification steps to enhance the realism of the generated image. As described in Section 4, $r_m$ is set to 6.0, and $r_M$ is set to 12.0. $\delta_1$ is sampled from a uniform distribution, ranging from 0.1 to 0.5. $\delta_{i+1}$ is also sampled from this distribution and added to $\delta_i$. The setting of $\gamma$ is the same as $\delta$. Table 5 showcases the typical results when SCA is introduced (denoted as CDCR+).

**Table 5.** Effect of SCA on results at ultra-high magnifications. The optimal PSNR values for each scale factor are shown in **bold**.

| Dataset | Method | Settings | Integer-Factor | | | | | Decimal-Factor | |
|---|---|---|---|---|---|---|---|---|---|
| | | | ×6 | ×8 | ×12 | ×16 | ×20 | ×9.7 | ×17.6 |
| RSC11 [47] | CDCR | base | 25.54 | 24.39 | 23.05 | 22.25 | 21.68 | 23.72 | 22.01 |
| | CDCR+ | w/ SCA | **25.56** | **24.42** | **23.08** | **22.29** | **21.72** | **23.75** | **22.05** |
| RSSCN7 [48] | CDCR | base | 25.74 | 24.75 | 23.59 | 22.88 | 22.43 | 24.17 | 22.73 |
| | CDCR+ | w/ SCA | **25.75** | **24.77** | **23.62** | **22.91** | **22.46** | **24.19** | **22.76** |
| WHU-RS19 [49] | CDCR | base | 27.39 | 25.95 | 24.31 | 23.40 | 22.77 | 25.14 | 23.12 |
| | CDCR+ | w/ SCA | **27.41** | **25.98** | **24.35** | **23.44** | **22.81** | **25.17** | **23.16** |

The utilization of the SCA method is imperative in cases of ultra-high magnifications. To demonstrate the reliability brought by SCA, we display the visualization results in Figure 8.

The results in Figure 8 depict that SCA can decrease the presence of artifacts in the image and render a more realistic representation of the scene in some cases. It is worth noting that the complexity of SCA is high due to the large number of predictions involved. Therefore, we only display the results of CDCR+ in this section. The discussion of complexity can be found in Section 5.5.

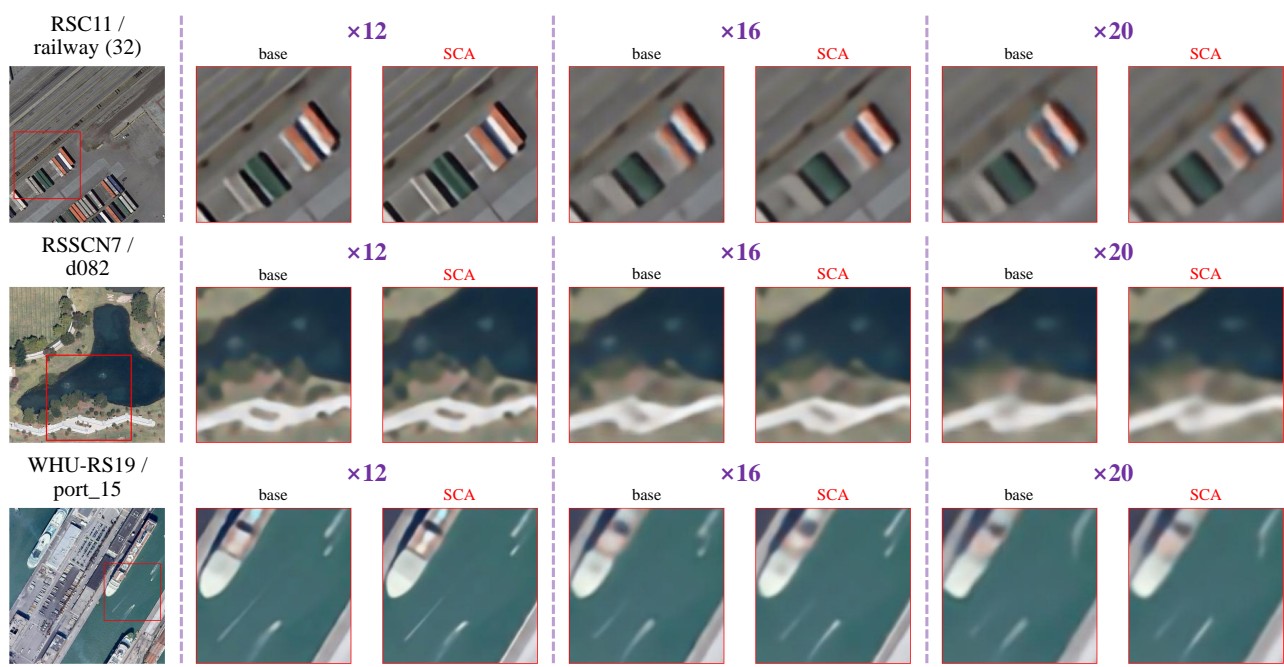

**Figure 8.** Enhancement brought about by SCA at scale factors of 12, 16, and 20.

### 5.5. Complexity Analysis

5.5.1. Experimental Environment

The experiments are conducted by a server cluster with a 64-bit Linux operating system. The hardware includes Tesla V100 GPU (32 GB memory) and Intel(R) Xeon(R) Gold 6230 CPU @ 2.10 GHz.

5.5.2. Complexity of Representative Methods

The complexity of each method is evaluated using the RSC11 test set and a scale factor of 9.7. The evaluation is performed by measuring the #FLOPs (G), #Params (M), and Inference Time (s) of each method. The results are presented in Table 6, where #FLOPs (G) denotes the average computation required for each image, #Params (M) represents the

number of model parameters, and Inference Time (s) signifies the average time required for inference on a single image. On the one hand, CDCR exhibits low computational overheads and a modest number of parameters. On the other hand, CDCR+ requires the prediction of multiple magnification steps, leading to a notable increase in the inference time. As such, CDCR+ is more suitable for scenarios in remote sensing where accuracy is paramount rather than computational efficiency.

**Table 6.** The statistics of FLOPs, Params, and Inference Time for different methods. #FLOPs (G) indicates the computation amount for a single image during the inference phase. #Params (M) indicates the number of model parameters, and Inference Time (s) indicates the average time taken to infer an image.

| Method | #FLOPs (G) | #Params (M) | Inference Time (s) |
|---|---|---|---|
| Meta-SR [1] | 3.83 | 1.67 | 0.076 |
| LIIF [2] | 5.99 | 1.57 | 0.234 |
| ArbSR [39] | 2.90 | 1.23 | 0.079 |
| A-LIIF [35] | 6.31 | 1.67 | 0.566 |
| CDCR (ours) | 6.05 | 1.58 | 0.270 |
| CDCR+ (ours) | - | - | 2.979 |

## 6. Conclusions

This paper proposes a novel image representation method, i.e., the combined discrete and continuous representation (CDCR), to address the challenging problem of reconstructing satellite images at arbitrary resolutions. As a plug-in method, CDCR can be integrated into existing super-resolution frameworks, enabling the generation of images at any desired resolution. Our CDCR combines the advantages of continuous representation (CR) and discrete representation (DR). On the one hand, the CR-based dense prediction ensures the generalization ability of the model, while on the other hand, the DR-based resolution-specific refinement with modulated modules improves high-frequency information in generated images and mitigate over-smoothing issues that may arise from CR. Additionally, this paper introduces a scale cumulative ascent (SCA) method during the inference phase to tackle the underfitting problem at ultra-high magnifications for the first time. The SCA method requires a large amount of inference time to produce more accurate images, which is crucial for remote sensing scenes with high accuracy requirements. To the best of our knowledge, this is the first work to systematically categorize and compare a majority of scale-arbitrary super-resolution methods in remote sensing scenes. As a general model, it may be considered to decrease the number of hyperparameters in CDCR to mitigate the potential impact of excessive manual design on the generated results. In the future, our focus will be on improving the efficiency of image representation methods, including enhancements to the encoder, to drive further advancements in super-resolution techniques in remote sensing.

**Author Contributions:** Conceptualization, T.A. and C.H.; methodology, T.A.; software, T.A. and C.H.; validation, T.A., C.H. and S.X.; formal analysis, S.X.; investigation, S.X.; resources, C.P.; data curation, T.A.; writing—original draft preparation, T.A.; writing—review and editing, C.H., S.X. and C.P.; visualization, T.A.; supervision, C.H. and C.P.; project administration, C.P.; funding acquisition, C.H. All authors have read and agreed to the published version of the manuscript.

**Funding:** This research is supported by National Natural Science Foundation of China (Grant No. 62071466), Fund of National Key Laboratory of Science and Technology on Remote Sensing Information and Imagery Analysis, Beijing Research Institute of Uranium Geology (Grant No. 6142A010402) and Guangxi Natural Science Foundation (Grand No. 2018GXNSFBA281086).

**Data Availability Statement:** Four public datasets (i.e., AID, RSC11, RSSCN7, and WHU-RS19) were included in this study. The data of AID were downloaded from the official website: https://captain-whu.github.io/AID/ (accessed on 12 August 2016). The data of RSC11 were obtained from the following URL provided by the author Lijun Zhao: https://pan.baidu.com/s/1mhagndY (accessed on 15 January 2015). The data of RSSCN7 were downloaded from the URL provided by

the author Qin Zou: https://pan.baidu.com/s/1slSn6Vz (accessed on 9 May 2016). The data of WHU-RS19 were downloaded from the official website: https://captain-whu.github.io/BED4RS/ (accessed on 7 July 2010).

**Conflicts of Interest:** The authors declare no conflict of interest.

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
