# Peer review of "Combining Discrete and Continuous Representation: Scale-Arbitrary Super-Resolution for Satellite Images"

_remotesensing, doi:10.3390/rs15071827_

Round 1

Reviewer 1 Report

This paper proposes a novel architecture for arbitrary scale single image super-resolution, by combining architectures proposed in [1] and [2], achieving better results than both original architectures and other state of the art architectures on 3 Google Earth based images dataset used for testing. The paper is reasonably well structured and well written, with a few clarifications and enhancements required locally (see bellow). In addition to local edits, there are howeve rmajor concerns that should be addressed for the audience of MDPI Remote Sensing, I therefore would recommend a major revision.

General concerns

The only remote sensing related elements  in this article are the training and testing dataset that are used. Methodology is not specific to remote sensing (which is fine) but the authors show a lack of knowledge of the remote sensing field, and several biases toward the Google Earth based dataset that have been used, which makes the paper less interesting for the MDPI Remote Sensing audience than it could be. This includes:
- Repeatedly  stating that images have 3 channels (RGB), while remote sensing sensors (either airborne or satellite) most often have 4 bands or more (for instance Sentinel-2 has 12 spectral bands with 3 different spatial resolutions)
- Confusing image size with spatial resolution, the latter generally refers to the size of ground projected pixel, and not the amount of pixel in an image (see for instance p.9 l. 290)
- Failing to relate to standard signal and image processing operators that are well-known in the community: for instance the intricate explanation of eq (8) is actually a standard bi-linear interpolation of the feature map (https://en.wikipedia.org/wiki/Bilinear_interpolation)
- Completely ignoring the concept of Modulation Transfer Function (https://en.wikipedia.org/wiki/Optical_transfer_function) of sensors, which is key to relate spatial resolution and level of sharpness / aliasing in remote sensing (and other vision fields as well, but in remote sensing the spatial resolution is usually constant for a given sensor, which makes MTF more prominent)

While for most part of the paper, this should be a matter of changes to formulations, for the experiments part however it is more than that: downgrading high resolution images with plain bicubic as stated in l. 301 will only result in LR images with a high MTF constant value at Nyquist Rate, and consequently sharp images with a lot of aliasing, while in remote sensing this MTF (and thus the trade-off between blur and aliasing) usually varies from one sensor to another, but is more on the blur side than on the sharp side, to avoid aliasing artifacts. Moreover, down-sampling images effectively reduce the noise in the image (averaging 4 pixels reduces SNR by a sqrt(2) factor or so), and therefore the higher the scale factor for down-sampling, the smaller the image noise is. This means that the simulated LR images are more sharp and far less noisy than any real remote sensing image you may apply the trained network to in real life. Therefore, the performances displayed in results table are probably over-estimated by a large amount for real-life applications.

As far as I can tell, all dataset are somehow extracted from Google Earth, which is another problem : images in Google Earth are processed for nice rendering for users, and they might be significantly different from real remote sensing images. For instance, their pixel depth is converted from source depth that can be as large as 12 bits to 8 bits, with various possibly non-linear scaling of the dynamic. It is possible that the processing also implies denoising, sharpening and other kind of image restoration (which are hidden in Google black box). This should be clearly stated in the paper.

Another limitation of those Google Earth based dataset is that they are composed of mixed actual spatial resolutions, depending on the source sensor and the level of zooming set during capture. If authors of AID dataset claim that actual spatial resolution ranges from 8m to 0.5m, it is unclear whether this information is available for the testing dataset, and available on a per image basis. This matters because it makes it very difficult to tell if the method is able to improve resolution passed the initial (maximum) HR resolution of the training set used for simulating LR images, which seems to be 0.5m. Landscapes are not scale-invariant, and it is a very different business to add 12.5cm details to a 50 cm image than to add 50cm details to a 2m image. Authors should find a way to demonstrate this ability in order to prove that the method is valuable for real life applications.

I would strongly recommend finding a real remote sensing dataset to complement the study, which may include for instance :
https://arxiv.org/abs/2207.06418
https://www.mdpi.com/2306-5729/7/7/96
https://github.com/ozgunhaznedar/Super-Resolution-for-Satellite-Imagery

Local edits

- p. 1, l. 19 "[...] or a reduction in sampling resolution" : this is simply not true. The major driver for spatial resolution of satellite sensors is physic of the sensor : size of detectors, signal to noise ratio in selected bandwidths, optical and motion blur, and satellite altitude.
- eq (3) p. 5 : shouldn't r be part of Phi or W ?
- p. 5, l. 184 : define A
- p. 7, section 3.3 : this whole section needs to be clarified. Figure 5 is most certainly artistic, but not very informative : we can not see that you are actually sequentially apply the model with resolution factor that combines into the target resolution factor. It may even make the reader think that you are simulating different viewpoints.
- Results table 1 - 5 : given that PSNR values are so close one to each other, it is not that obvious that your method significantly outperform others. Maybe you should consider including other metrics to highlight it. There is at least one error in bold best values (table 3, last column, CDCR-c1 is better than CDCR-c9 but the latter is in bold).

Reviewer 2 Report

In this work, the authors propose a novel image representation method called CDCR, which combines discrete representation (DR) and continuous representation (CR) of images. The CDCR methods can serve as a plug-in module to help image super-resolution frameworks accommodate arbitrary scale factors.

They applied the method to various remote sensing datasets, such as RSC11, RSSCN7 and WHU-RS19.

The paper is interesting and its final goal is an actual research topic.

Anyway, in my opinion, the introduction could be improved with more background and references to recent research in a similar/same topic.

Author Response

Thank you for your review. We have included more research background and introduced additional references in Section 2.3 of the manuscript.

Reviewer 3 Report

This paper proposes a scale-arbitrary super-resolution for satellite images by combining discrete and continuous representation. The proposed method consists of two components: a CR-based dense prediction that gathers more available information and a DR-based resolution-specific refinement that adjusts the predicted values of local pixels. Extensive experiments on multiple remote sensing datasets validate the effectiveness of the proposed method. The proposed idea is interesing. The experimental results are adequate. My additional comments are as follows.

(1) The limitations and the failure cases should be discussed.

(2) The authors should provide the detailed experimental parameters.

(3) The English should be polished.

Author Response

Thank you for reviewing our work. One limitation of this study is that it involves more hyperparameters than usual. As a universal pipeline, it is crucial to further reduce hyperparameters to ensure the model's generalizability. We have added this limitation to the Conclusions.

The detailed experimental parameters are described in Section 5.2, while the SCA configuration is provided in Section 5.4.4. We provide the code at https://github.com/Suanmd/CDCR/, which is publicly available for readers interested in more training and testing details.